# GcvB Regulon Revealed by Transcriptomic and Proteomic Analysis in *Vibrio alginolyticus*

**DOI:** 10.3390/ijms23169399

**Published:** 2022-08-20

**Authors:** Bing Liu, Jianxiang Fang, Huizhen Chen, Yuehong Sun, Shan Yang, Qian Gao, Ying Zhang, Chang Chen

**Affiliations:** 1CAS Key Laboratory of Tropical Marine Bio-Resources and Ecology (LMB), Guangdong Provincial Key Laboratory of Applied Marine Biology (LAMB), South China Sea Institute of Oceanology, Chinese Academy of Sciences, Guangzhou 510301, China; 2College of Earth and Planetary Sciences, University of Chinese Academy of Sciences, Beijing 101408, China; 3School of Environment, South China Normal University, Guangzhou 510000, China; 4College of Marine Sciences, South China Agricultural University, Guangzhou 510642, China; 5Xisha Marine Environmental National Observation and Research Station, Sansha 573199, China

**Keywords:** *Vibrio alginolyticus*, GcvB, amino acid metabolism, T3SS, Hfq

## Abstract

*Vibrio alginolyticus* is a widely distributed marine bacterium that is a threat to the aquaculture industry as well as human health. Evidence has revealed critical roles for small RNAs (sRNAs) in bacterial physiology and cellular processes by modulating gene expression post-transcriptionally. GcvB is one of the most conserved sRNAs that is regarded as the master regulator of amino acid uptake and metabolism in a wide range of Gram-negative bacteria. However, little information about GcvB-mediated regulation in *V. alginolyticus* is available. Here we first characterized GcvB in *V. alginolyticus* ZJ-T and determined its regulon by integrated transcriptome and quantitative proteome analysis. Transcriptome analysis revealed 40 genes differentially expressed (DEGs) between wild-type ZJ-T and *gcvB* mutant ZJ-T-Δ*gcvB*, while proteome analysis identified 50 differentially expressed proteins (DEPs) between them, but only 4 of them displayed transcriptional differences, indicating that most DEPs are the result of post-transcriptional regulation of *gcvB*. Among the differently expressed proteins, 21 are supposed to be involved in amino acid biosynthesis and transport, and 11 are associated with type three secretion system (T3SS), suggesting that GcvB may play a role in the virulence besides amino acid metabolism. RNA-EMSA showed that Hfq binds to GcvB, which promotes its stability.

## 1. Introduction

GcvB, originally identified in *E. coli* as part of the glycine cleavage system [1], is one of the most conserved small RNA (sRNA) in a wide spectrum of Gram-negative bacteria such as *Enterobacteriaceae**,*
*Actinobacillus*, *Pasteurella*, *Photorhabdus*, and *Vibrio* [2,3,4,5]. It is considered as the master sRNA regulator of amino acid metabolism. Miyakoshi et al. (2021) summarized that GcvB directly regulates more than 50 direct target genes in *E. coli* and *Salmonella,* which includes amino acid metabolism, ABC transporter and permease, antiporter, carbon metabolism, membrane integrity, RNA metabolism, and transcriptional regulator [6]. In addition, GcvB modulates critical cellular processes such as growth ability [7], biofilm formation [8], two-component system [9], acid resistance [10], and oxidative stress response [11], sensitivity to aminoglycoside antibiotics [12] in *gamma-proteobacteria*.

Previous studies revealed that GcvB utilizes three conserved seed sequences, namely R1, R2, and R3 to regulate multiple target genes. The G/U-rich R1 region is capable of base-pairing interactions with vast majority of previously known targets [2,13,14,15,16,17]. The R3 seed sequence regulates several mRNAs including *phoP* and *lrp*, which encode global transcriptional regulators [9,18,19], as well as sRNA SroC [20]. Although the R2 sequence is highly conserved, it may only be utilized to repress *cycA* mRNA in *E. coli* and *Salmonella* [16,21]. Upon technical developments, new methodologies such as RNA-seq [3,7,11], RIL-seq (RNA interaction by ligation and sequencing) [22,23], CLASH (UV cross-linking, ligation, and sequencing of hybrids) [24], and MAPS (MS2-affinity purification coupled with RNA sequencing) [19] were performed to explore GcvB regulon, leading to quick expansion of GcvB sRNA–mRNA interactome data sets [6,19].

*Vibrio alginolyticus* is a common Gram-negative opportunistic pathogen widely distributed in the marine and estuarine environments where a variety of carbon and nitrogen sources are supplied, and poses a potential threat to marine animals and human [25,26,27,28,29,30,31,32,33]. The RNA binding proteins, Hfq and CsrA that play central roles in sRNA functioning, have been shown to be critical for the fast growth and highly effective metabolism of carbohydrates and amino acids of *V. alginolyticus* previously [34,35], indicating that sRNAs may be the key elements in the regulation of metabolism in response to the changing environments.

GcvB function has primarily been evaluated in the family of *Enterobacteriaceae*, which leaves a question of what role it may play in other bacteria that live in habitats different from those of *Enterobacteriaceae*. Recently, a study on *Pasteurella multocida* has shown that GcvB functions as an amino acid metabolism controller as in other bacteria, while its regulatory targets are very different [3]. Previously, Silveira et al. [5] have reported that GcvB homolog is widely distributed in species of the *Vibrionacea* family. However, its physiological role and regulatory targets remain unknown. To address this issue, we characterized *gcvB* and identified its regulon by integrating the high-resolution RNA-seq and DIA assays in *V. alginolyticus* ZJ-T. It is the first report to reveal the regulatory role of GcvB in the *Vibrionacea* family, which may also shed light on the functional and evolutionary diversity of this conserved sRNA in different bacteria.

## 2. Results

### 2.1. Bioinformatic Analysis of GcvB Sequence of V. alginolyticus and Its Expression in LBS

In a previous study, trans-encoded regulatory sRNAs were identified in the genome of *Vibrio alginolyticus* ZJ-T [36]. GcvB locates in the intergenic region of chromosome II, between BAU10_02485 (encoding tRNA 4-thiouridine (8) synthase ThiI) and BAU10_02490 (encoding transcriptional regulator GcvA) (Figure 1A). In other bacteria, *gcvB* and *gcvA* are always oriented together and transcribed divergently, but the downstream genes are variable. The *gcvB* gene contains a non-coding sequence of 211 nucleotides, which shows high similarity to the GcvB ortholog of the organisms from the families of *Enterobacteriaceae*, *Vibrionaceae*, and *Pasteurellaceae* (Figure 1C). However, except R1 and R2 sequences that are common to all GcvB sRNAs, the conserved R3 sequence in *Enterobacteriaceae* is not present across those of *Vibrionaceae* (Figure 1D).

To determine how GcvB is expressed in *V. alginolyticus*, we analyzed the transcriptome (RNA-seq) data generated from the RNA isolated from the cells at the early exponential phase (OD_600nm_ = 0.5). The average reads per kilobase per million mapped reads (RPKM) is 7697, indicating a strong expression of *gcvB*. Quantitative RT-PCR showed that *gcvB* expression increased with growth: at stationary phase (OD_600nm_ = 5.0), the abundance of *gcvB* transcripts increased by six-fold compared to early exponential phase (Figure 1B), which is in contrary with the reports of other bacteria [2,13,19].

### 2.2. Construction of the Mutant and Complementary Strains and Measurement of Their Growth Ability under Different Conditions

In order to examine the role of GcvB in *V. alginolyticus*, we constructed an in-frame deletion of *gcvB* in the ZJ-T, named ZJ-T-△*gcvB*, and accordingly a complementary strain ZJ_T-△*gcvB*^+^ that harbored a pMMB207 plasmid carrying the fragment of *gcvB* driven by the promoter of P_tet_ in the plasmid. The *gcvB* expression in the strains were examined by qRT-PCR. The result confirmed the mutant totally lost its transcription, but the complementary strain has only partially restored the expression of *gcvB* (50% compared to the wild type) (Figure 2), which may be due to the difference in promoters.

The growth of ZJ-T, ZJ-T-Δ*gcvB*, and ZJ-T-Δ*gcvB*^+^ were measured under several conditions. As shown in Figure 3, there is no significant difference among them when the cells were grown in rich medium LBS (Figure 3A) and minimum media M63 supplemented with NH_4_^+^((NH_4_)_2_SO_4_) plus D-glucose, indicating that deletion of *gcvB* did not affect the uptake and metabolism of carbon and nitrogen sources, nor the amino acid biosynthesis processes from de novo. To investigate the impact of GcvB on the uptake and catabolism of specific amino acids, bacterial growth was measured in M63 supplemented with L-alanine, branched-chain amino acids (L-isoleucine, L-leucine and L-valine), L-aspartic acid, L-arginine, L-threonine, and L-serine. As shown in Figure 3, all of the three strains showed similar growth curves except that when they were grown in M63 plus L-alanine or M63 plus L-alanine plus D-glucose, ZJ-T-Δ*gcvB* had a much longer lag phase (>8 h) compared to ZJ-T and ZJ-T-Δ*gcvB*^+^ partially restored this phenotype. This suggests that *gcvB* may be involved in alanine uptake and/or catabolism.

### 2.3. Integrative Transcriptome and Proteome Analysis of the Wild Type Strain and GcvB Mutant

To identify the regulon of *gcvB* of *V. alginolyticus*, whole-transcriptome RNA sequencing (RNA-seq) and DIA (data-independent acquisition) quantitative proteomics were performed to analyze the transcriptome and proteome of ZJ-T and ZJ-T-△*gcvB*. The RNA and protein samples were prepared from cells harvested at early exponential phase (OD_600_ = 0.5).

According to the data of transcriptome, on average, 21.2 and 21.4 million high-quality 150 bp paired-end clean reads of ZJ-T and ZJ-T-Δ*gcvB* group respectively were mapped to the genome of *V. alginolyticus* ZJ-T. Quality analysis of the transcriptome showed an average Q20 of 98.43% and 98.46% and Q30 of 95.03% and 95.10% for ZJ-T and ZJ-T-Δ*gcvB*, respectively. The average map rates were 95.99% for ZJ-T and 96.29% for ZJ-T-△*gcvB*, respectively (Appendix A). The differentially expressed genes (DEGs) were identified with the absolute value of fold changes of ZJ-T/ZJ-T-Δ*gcvB* |FC| ≥ 2 and a false discovery rate-adjusted *p* (q value) < 0.05. Protein identification and quantification were done by nano-HPLC-MS/MS. A total of 37,861 peptides that are matched with 3276 proteins were detected, accounting for 72.8% of the entire encoded proteins in the genome of ZJ-T, of which 606 proteins matched with less than three peptides. The differentially expressed proteins (DEPs) were identified with FDR< 0.05 and |FC| ≥ 1.5.

A total of 40 DEGs (differentially expressed genes) were identified (Appendix A). Compared to the wild-type strain, ZJ-T-△*gcvB* showed 27 up-regulated and 13 down-regulated genes, respectively (Figure 4A,B). However, qRT-PCR confirmed that only 18 of them showed similar expression differences (Data not shown). The DEGs were enriched in 2 KEGG pathways, namely “histidine metabolism” and “arginine biosynthesis” with the Q value < 0.05 (Figure 4C). For proteome analysis, 50 proteins displayed significant differences after inactivation of *gcvB*, 36 of which showed increased production and 14 of which were down-regulated (Figure 5A,B). The DEPs were enriched in three primary KEGG pathways, namely “sulfur metabolism”, “branched chain amino acid biosynthesis”, and “type three secretion system” (Figure 5C).

Integrative analysis of transcriptome and proteome data showed only four genes are presented both in DEGs and DEPs (Figure 6A), namely *mtlA* (encoding PTS mannitol transporter subunit II), *mtlD* (encoding mannitol-1-phosphate 5-dehydrogenase), *HI_1246* (encoding LTA synthase family protein), and *cysM* (encoding cysteine synthase) (Appendix A.). All of them were up-regulated similarly in *gcvB* mutant at both mRNA and protein levels. As many as 46 out of 50 DEPs (92%) showed no difference in transcription, indicating the majority of different expression of DEPs is the consequence of post-transcriptional regulation by *gcvB*. However, there are 36 DEGs not presented in the DEPs dataset too. Additionally, the 20 DEPs, with a reliable STRING score (>700), were shown to be involved in a PPI network based on the STRING database (Figure 6B), indicating they interacted with each other.

### 2.4. Identification of GcvB Regulon and Its Possible Physiological Role in V. alginolyticus ZJ-T

Based on the primary KEGG pathways they are involved in, the DEGs and DEPs are categorized, which may help to understand the physiological role of GcvB in *V. alginolyticus*.

#### 2.4.1. Valine/Leucine/Isoleucine Biosynthetic Pathway

Like *E. coli*, the genome of *V. alginolyticus* ZJ-T contains a whole set of genes for BCAAs biosynthesis from de novo, which consists of four operons, namely *ilvGMEDA*, *ilvBN*, *ilvIH,* and *leuABCD*. They are located in different regions of the genome, where *ilvGM*, *ilvBN,* and *ilvIH* encode three isozymes of acetolactate synthase (AHAS), a key enzyme for branched-chain amino acid synthesis, respectively, but *ilvGM* plays a dominant role in most conditions [37]. Proteomic analysis showed that GcvB inhibited the expression of *ilvA*, *ilvG* and *leuB*, *leuC*, *leuD,* as well as *brnQ* that encodes BCAAs transporter. Although they did not display significant difference in the transcriptome analysis, qRT-PCR showed about slight but significant increases of *ilvC*, *ilvE,* and *ilvD* expression in ZJ-T-△*gcvB* compared to ZJ-T (Figure 7A). To verify whether GcvB regulates these candidates at the post-transcriptional level, translational fusion of *ilv*G to *gfp* gene was carried out and determined by Western blot. As shown in the Figure 7B, it was not expressed in the early log phase of both wild-type and *gcvB* mutant, but its expression increased with growth from the mid-log phase onward. Compared with ZJ-T, IlvG showed a significant up-regulation during mid-log phase growth in ZJ-T-Δ*gcvB*, but the expression was the same when the cells reached to the stationary phase, indicating that the repression of *ilvG* by GcvB occurs during exponential growth.

#### 2.4.2. Sulfur and Cysteine Biosynthesis Metabolism

Sulfur is an essential element for life and the metabolism of organic sulfur compounds plays an important role in the global sulfur cycle. The *Vibrio* family can efficiently uptake sulfate from seawater and convert it into cysteine and methionine, a process catalyzed by proteins encoded by 19 genes on the genome [38]. Proteomic data showed an over-representing of genes (*cysI*, *cysJ*, *cysH*, *cysN*, *cysD,* and *cysK*) in sulfur metabolism and cysteine synthesis pathway (*p* = 0.00019). They are located on three different operons, namely *cysGDN*, *cysZK*, and *cysJIH*. In the *gcvB* mutant, the quantity of their proteins was reduced by 2–5-fold compared to the wild type, but the abundance of their transcripts quantified by RNA-seq and qRT-PCR did not differ (Figure 8A, Appendix A), indicating a positive regulation of their expression by GcvB from post-transcription, contrary to the common pattern of regulation of GcvB reported so far. To verify this regulation, we constructed the translational fusion of *cysK* and *cysN* to *gfp*, and measured their expression in LBS along the growth. As shown in the Figure 8, both of them are not expressed in the early log phase, but are strongly expressed from the mid-log phase and remain high in the stationary phase. Compared to ZJ-T, *cysN* is not expressed in ZJ-T-Δ*gcvB* during the growth and *cysK* is significantly down-regulated.

However, in the transcriptome data of ZJ-T-Δ*gcvB*, *cysM* encoding cysteine synthase was up-regulated by 5.56-fold compared to that of ZJ-T, which was verified by qRT-PCR (Figure 8A). Meanwhile, its protein was increased approximately five-fold, despite an FDR value of more than 0.05. It suggests that *gcvB* may indirectly regulate the expression of *cysM* by repressing an unknown transcriptional factor that is required for *cysM* transcription (Figure 8B). These data refer that GcvB is likely to be involved in the regulation of sulfur metabolism and cysteine biosynthesis pathway by at least two different mechanisms in *V. alginolyticus*.

In addition, transcriptomic data showed that the *hutI*, *hutG,* and *hutU* genes encoding proteins that catalyze the conversion of histidine to glutamate were more than two-fold upregulated in the *gcvB* mutant (Figure 9).

#### 2.4.3. ABC Transporters

Previous work in *E. coli* and *Salmonella* revealed that GcvB represses multiple target mRNAs, most of which encode amino acid uptake systems relevant for the utilization of external nitrogen sources. In this study, the *artP* (encoding arginine ABC transporter ATP-binding protein) was upregulated by 5.52-fold in mRNA level while *aapJ* (encoding amino acid ABC transporter, periplasmic amino acid-binding protein) and *metN* (encoding methionine ABC transporter ATP-binding protein) were upregulated by 2.69 and 1.57-fold respectively in their protein levels. It is noteworthy of the general L-amino acid permease (Aap) that is encoded by an operon containing four genes *aapJQMP*. AapJ is a periplasmic binding protein that has a broad ligand specificity which is required for transport of all solutes [39,40]. It indicated that GcvB negatively regulates the uptake of amino acids in *V. alginolyticus*.

#### 2.4.4. Bacterial Secretion Systems

Bacterial secretion systems are widespread in bacteria allowing them to infect eukaryotic cells or compete with non-akin bacteria [41]. Many Gram-negative pathogens employ T3SS to translocate effector proteins into eukaryotic host cells, which is important for bacterial survival and virulence [42,43] while T6SS is important for bacterial competition through contact-dependent killing of competitors [44,45,46].

In this study, the proteomic data showed T3SS-associated genes including *yscB* (encoding T3SS chaperone), BAU10_07880 (encoding T3SS chaperone), *yscP* (encoding T3SS needle length determinant), *yscF* (encoding T3SS export protein)*, exsE2*(encoding T3SS regulator), *yscV* (encoding T3SS protein V) were up-regulated by approximately two-fold in *gcvB* mutant compared to the wild type, although no difference in their transcripts. However, T6SS-associated genes including *tssC* (type VI secretion system contractile sheath large subunit), *tssB* (type VI secretion system contractile sheath small subunit), *tagH* (type VI secretion system-associated FHA domain protein) showed significant down-regulated expression in their mRNA abundance, but no significant difference in their proteins. Therefore, it may suggest that *gcvB* represses T3SS but activates T6SS of *V. alginolyticus*.

### 2.5. Effects of Hfq on GcvB

Hfq is an RNA chaperone that assists interactions between sRNA and its targets and or enhances stabilities of many sRNAs [47]. To investigate the involvement of Hfq in GcvB regulation, qRT-PCR was used to quantify its expression in LBS medium between the wild type ZJ-T and an *hfq* mutant ZJ-T-Δ*hfq*. As shown in Figure 10A. GcvB was down-regulated in early exponential and stationary phase in ZJ-T-Δ*hfq* but no significant difference in mid-exponential period compared to the wildtype was observed, indicating that Hfq positively regulates the expression of GcvB. The stability of GcvB RNA was determined by measuring its half-life (Figure 10B). The result showed that it was beyond 15 min in the wild-type strain but less than 4 min in the *hfq* deletion strain, indicating that Hfq promotes GcvB RNA stability. To examine if Hfq binds directly to GcvB, we performed an RNA electrophoretic mobility shift assay (REMSA) with purified Hfq protein and a biotinylated GcvB RNA oligonucleotide that contains 211 nucleotides of the entire transcript. The result showed Hfq binds to GcvB from 0.15 μg (Hfq = 1.40318 × 10^−11^ mol; GcvB = 2.9567 × 10^−13^ mol) to 1.05 μg and there was no change in the mobility of Hfq at a concentration over 1.05 μg (Hfq = 9.82226 × 10^−11^ mol; GcvB = 2.9567 × 10^−13^ mol) (Figure 10C), indicating that the concentration at which Hfq protein binds to GcvB reaches saturation at 1.05 μg. To verify that Hfq binds to GcvB specifically, after the non-biotinylated GcvB probes were added to the last three groups (groups 12, 13, 14), the labeled GcvB was competitively eluted and the band reappeared below, indicating the binding of GcvB is specific.

## 3. Discussion

In this study, we characterized the physiological role of GcvB of *V. alginolyticus* ZJ-T, and identified its regulon. Deletion of *gcvB* has been reported to reduce generation time of *Yersinia pestis* [4], but no effect was seen on the growth rate of *E. coli* [1]. The *gcvB* gene encodes a small untranslated RNA, involved in the expression of dipeptide and oligopeptide transport systems in *Escherichia coli*. In *V. alginolyticus*, deletion of *gcvB* resulted in no difference in growth rate with the wild type when the cells grew either in rich medium (LBS) or defined media, except when alanine was used as the sole carbon/nitrogen source. It suggests that GcvB is likely to affect the uptake and/or the initiation degradation of alanine, but which target genes are affected and their mechanisms need to be determined by further experiments.

By comparing the transcriptome and proteome data between the *gcvB* knockout strain and its wild-type parental strain that grows at early exponential phase, we found that GcvB affects the expression of <1% (0.86%) of the *V. alginolyticus* transcriptome and 1.52% proteome. The regulatory roles of GcvB identified in this analysis are summarized in Figure 11. Transcriptomics-based studies in *Enterobacteriaceae*, such as *E. coli* and *Salmonella*, have shown that GcvB directly or indirectly regulates the expression of about 1–2% of total genes, or about 50–100 genes [20] of the genome. The number of experimentally confirmed target genes has exceeded 50. Among them, more than 30 genes encode proteins for amino acid transport and metabolism. In this study, of the genes that showed either increased (36) or decreased (14) protein levels in the GcvB-deficient strains compared to the wild type, 18 were predicted to be involved in amino acid biosynthesis and transport, suggesting that GcvB acts primarily to repress the biosynthesis and transport of amino acids during the early growth stages in *V. alginolyticus*, likely as a means to conserve energy when nutrients are abundant [3]. In *E. coli* and *S. Typhimurium*, the majority of GcvB regulon is associated with amino acid transporters (>60% of GcvB targets) [16]. But in *V. alginolyticus*, only 3 out of 50 genes are responsible for amino acid transport, while 30% of the regulon is involved in the biosynthesis of amino acids. Interestingly, only three DEPs in *V. alginolyticus* are also presented in the *gcvB* regulon of *E. coli* and *S. Typhimurium*. Gulliver et al. recently identified the GcvB regulon in *Pasteurella multocida* by quantitative proteome analysis, showing that most part of the regulon is not shared with those in *Enterobacteriaceae**,* although its major role is similar [3]. It may suggest that the function of *gcvB* is conserved across families to most extend, but its targets are diverse, which may result from the co-evolution consequences required for different bacteria surviving strategies.

Sulfur is an element essential for microbial life [48]. Seawater contains 27.7 mM of sulfate, which can be assimilated by various marine microorganisms, of which *Vibrio* is typical [49,50]. It was reported that CysB, an LysR family transcriptional regulator, is required for the transcription initiation of the multiple *cys* operons, but little was known about the regulatory mechanisms of sulfur metabolism in *Vibrio*. In this study, the amount of proteins such as CsyI, CsyJ, CsyN, CsyK, CsyH, and CsyD was decreased in *gcvB* mutant, while the abundances of their transcripts are not altered, referring that GcvB positively regulates their expression post-transcriptionally. This is in contrast to the most cases except that GcvB was reported to positively regulate RNase BN/Z by stabilizing its mRNA [51]. How *gcvB* positively regulates the expression of *cys* genes remains to be elucidated.

In addition to metabolism, we here first found that GcvB may be also involved in the virulence of *V. alginolyticus* by modulating a large part of T3SS gene expression post-transcriptionally. T3SS is an important virulence factor of *V. alginolyticus*, which induces apoptosis and autophagy of the host cells, so the virulence is greatly reduced in the absence of T3SS [52,53]. In *Pseudomonas aeruginosa*, sRNA 179 was reported to negatively regulate T3SS by repressing the Gac/Rsm signal transduction system that is required for the expression of T3SS regulon [54], but so far no study hints the link between GcvB and T3SS.

This study first identified and characterized the GcvB regulon in *V. alginolyticus* strain ZJ-T. Compared to previous studies in other bacteria, the sequences and primary roles of GcvB are well conserved, but its targets are different among the bacteria. Furthermore, we first found it may be also involved in cysteine biosynthesis and virulence, but the targets and mechanism need to be further revealed.

## 4. Materials and Methods

### 4.1. Bacterial Strains, Plasmids, and Media

All bacterial strains and plasmids used in this study are listed in Table 1. All strains were maintained at −80 °C in tryptic soy broth (TSB) (BD, New Jersey, USA) plus 25% glycerol. *V. alginolyticus* and derivatives were routinely cultured in TSB or lysogeny broth (LB) (VWR International, Radnor, PA, USA) plus 2.5% NaCl at 30 °C. *Escherichia coli* strains were cultured in LB medium supplemented with appropriate antibiotics at 37 °C. For the selection of transconjugants, TCBS medium (HuanKai, Guangzhou, China) was used with 5 μg/mL chloramphenicol (Cm) and 0.2% D-glucose. To select transconjugants that had undergone plasmid excision and allelic exchange, TCBS medium plus 0.2% arabinose plus 5 μg/mL chloramphenicol (Cm) or TCBS medium plus 0.2% arabinose alone was used to induce the *ccdB* gene and to select bacteria that had lost the inserted plasmid. Antibiotics were used at the following concentrations: chloramphenicol (Cm) at 5 μg/mL for *V. alginolyticus* and 20 μg/mL for *E. coli*; ampicillin (Amp) at 100 mg/mL for *E. coli*. When necessary, diaminopimelate (DAP) was added to the growth media at a final concentration of 0.3 mM.

### 4.2. Phylogenetic Tree and Sequence Analysis

The sequences were obtained from GenBank. The phylogenetic tree and sequence analysis were constructed based on the DNA difference with the ML (maximum likelihood) method with 300 bootstrap replicates using MEGA X (downloaded from http://www.megasoftware.net/, accessed on 6 April 2022). The tree was visualized via iTOL (iTOL: Interactive Tree of Life (https://itol.embl.de/), accessed on 6 April 2022)

### 4.3. Mutant and Complementary Strains Construction

The *gcvB* gene was deleted from *V. alginolyticus* ZJ-T as previously described [35] with slight modifications. In brief, upstream and downstream of the target gene *gcvB* were PCR-amplified with the primer pairs annotated *gcvB*-UP-F/R and *gcvB*-DOWN-F/R (Appendix A), and the vector fragment pSW7848 was PCR-amplified with the primer pair pSW7848-F/R (Appendix A). The recombinant suicide plasmid pSW7848-Δ*gcvB* was obtained by isothermal assembly and transformed into GEB883 cells (Table 1), which was then confirmed using the primers annotated Del-check-pSW7848-F/R. Conjugations and selection of mutants were carried out as previously described [35]. PCR and sequencing were used to check for the presence or absence of the target genes with the primer pair Δ*gcvB*-check-F/R.

The pMMB207 vector fragment and intact *gcvB* fragment were PCR-amplified with the primer pair pMMB207-F/R and *gcvB*-F/R (Appendix A), respectively, connected, and transformed into *E. coli* GEB883 cells. PCR and sequencing were used on colonies to check for the presence or absence of the target genes with the primer pair annotated com-pMMB207-F/R. The recombinant plasmid pMMB207-*gcvB* was transformed into *V. alginolyticus* mutant ZJ-T-Δ*gcvB* cells, resulting in the ZJ-T-△*gcvB^+^* strain (Table 1). All amplified DNA samples were sequenced to ensure no errors had occurred during amplification.

### 4.4. Growth Measurement

Growth measurements in rich medium LBS and different modified minimal medium M63 were carried out as previously described [35] with slight modification. To investigate the effect of amino acid(s) on growth, D-glucose and (NH_4_)_2_SO_4_ in M63 were left out and replaced by the amino acids L-alanine (150 mM), ILV (L-isoleucine, L-leucine and L-valine, 20 mM respectively), L-aspartic acid (50 mM), L-arginine (50 mM), L-threonine (50 mM), and L-serine (50 mM) as carbon and nitrogen sources. M63 medium plus 0.4% (*w*/*v*) D-glucose plus L-alanine (150 mM) was also used to investigate whether GcvB was involved in alanine metabolism. Minimal medium assays were carried out as previously described [35]. More than three replicates in each case and three repetitions of the experiment were carried out in these measurements.

### 4.5. RNA Extraction and Whole-Genome RNA-Sequencing

The experimental design comprised two groups: the wildtype strain ZJ-T and mutant syrain ZJ-T-Δ*gcvB* (n = 3 per group). LBS cultures from single colonies were grown overnight and then diluted 1:1000 in LBS medium and grown to the mid-log phase (OD_600nm_ ≈ 0.6), and 100 mL LBS cultures were collected. Total RNA was extracted by TRIzol-based method (Life Technologies, California, USA). RNA quality control was assessed on an Agilent 2100 Bioanalyzer (Agilent Technologies, Palo Alto, California, USA) and checked using RNase free agarose gel electrophoresis. The purified RNA was sent to Genedenovo Biotechnology Co., Ltd. (Guangzhou, China), where the RNA sample was assembled into a single ended RNA-Seq library and sequenced by Illumina Novaseq 6000 platform with pair-end 150 base reads. Raw data were filtered by the following standards and quality trimmed reads were mapped to the reference genome using Bowtie2 [60] (version 2.2.8) allowing no mismatches, and reads mapped to ribosome RNA were removed. Retained reads were aligned with the reference genome using Bowtie2 to identify known genes and calculated gene expression by RSEM [61].

The gene expression level was further normalized by using the fragments per kilobase of transcript per million (FPKM) mapped reads method to eliminate the influence of different gene lengths and amount of sequencing data on the calculation of gene expression. The edge R package (http://www.r-project.org/, accessed on 26 July 2021) was used to identify differentially expressed genes (DEGs) across samples with fold changes ≥2 and a false discovery rate-adjusted *p* (q value) < 0.05. DEGs were then subjected to an enrichment analysis of GO function and KEGG pathways, and q values < 0.05 were using as threshold.

### 4.6. Protein Extraction and Protein Digestion

Samples were collected as done in RNA-seq method, then were transferred into lysis buffer (2% SDS, 7 M urea, 1 mg/mL protease inhibitor cocktail), and homogenized for 5 min in ice using an ultrasonic homogenizer. The homogenate was centrifuged at 15,000 rpm for 15 min at 4 °C, and the supernatant was collected. BCA Protein Assay Kit was used to determine the protein concentration of the supernatant. About 50 μg proteins extracted from cells were suspended in 50 μL solution, reduced by adding 1 μL 1 M dithiothreitol at 55 °C for 1 h, alkylated by adding 5 μL 20 mM iodoacetamide in the dark at 37 °C for 1 h. Then the sample was precipitated using 300 μL prechilled acetone at −20 °C overnight. The precipitate was washed twice with cold acetone and then resuspended in 50 mM ammonium bicarbonate. Finally, the proteins were digested with sequence-grade modified trypsin (Promega, Madison, Wisconsin, USA) at a substrate/enzyme ratio of 50:1 (*w*/*w*) at 37 °C for 16 h.

### 4.7. High PH Reverse Phase Separation and DIA(Nano-HPLC-MS/MS Analysis)

The peptide mixture was re-dissolved in buffer A (buffer A: 20 mM ammonium format in water, pH10.0, adjusted with ammonium hydroxide), and then fractionated by high pH separation using Ultimate 3000 system (Thermo Fisher scientific, MA, USA) connected to a reverse phase column (XBridge C18 column, 4.6 mm × 250 mm, 5μm, (Waters Corporation, MA, USA). High pH separation was performed using a linear gradient, starting from 5% B to 45% B in 40 min (B: 20 mM ammonium format in 80% ACN, pH 10.0, adjusted with ammonium hydroxide). Ten fractions were collected; each fraction was dried in a vacuum concentrator for the next step.

The peptides were re-dissolved in 30 μL solvent A (A: 0.1% formic acid in water) and analyzed by on-line nanospray LC-MS/MS on an Orbitrap Fusion Lumos coupled to EASY-nLC 1200 system (Thermo Fisher Scientific, MA, USA). About 3 μL peptide sample was loaded onto the analytical column (Acclaim PepMap C18, 75 μm × 25 cm) with a 120-min gradient, from 5% to 35% B (B: 0.1% formic acid in ACN). The mass spectrometer was run under data independent acquisition mode, and automatically switched between MS and MS/MS mode. DIA was performed with variable Isolation window, and each window overlapped 1 *m*/*z*, and the window number was 60.

### 4.8. Protein Functional Annotation, Enrichment Analysis, and PPI Network Construction and Analysis

Raw data of DIA were processed and analyzed by Spectronaut X (Biognosys AG, Schlieren, Switzerland) with default parameters. Retention time prediction type was set to dynamic iRT. Data extraction was determined by Spectronaut X based on the extensive mass calibration. Spectronaut Pulsar X determined the ideal extraction window dynamically depending on iRT calibration and gradient stability. Qvalue (FDR) cutoff on precursor and protein level was applied 1%. Decoy generation was set to mutated, which was similar to scrambled but only applies a random number of AA position swamps (min = 2, max = length/2). All selected precursors passing the filters were used for quantification. The average top three filtered peptides which passed the 1% Qvalue cutoff were used to calculate the major group quantities. After Student’s *t*-Test, different expressed proteins were filtered if their Qvalue was 0.58.

Proteins were annotated against GO, KEGG, and COG/KOG database to obtain their functions. Significant GO functions and pathways were examined within differentially expressed proteins with q value ≤ 0.05. For PPI network construction and analysis, STRING (https://string-db.org/, accessed on 26 July 2021) database was utilized to create the PPI networks [62]. Further information on the possible function of differentially expressed proteins was predicted on potential PPIs using Cytoscape software [63] to identify and visualize potential PPIs.

### 4.9. Quantitative Reverse Transcription PCR (qRT-PCR) Analysis

qRT-PCR analysis were carried out to verified gene expression as previously described [35]. The relative expression of genes was detected by qPCR using gene-specific primers (Appendix A), and 16s rDNA was used as an internal reference. Relative levels were calculated using the threshold cycle (ΔΔCT) method [64] and normalized to the wild type ZJ-T value. Measurements were done in triplicate. Statistical significance was determined by the Student’s *t*-Test (ns *p* > 0.05, * *p* < 0.05, ** *p* < 0.01).

### 4.10. Translational Fusion

To create the translational fusion of target genes, PCR fragment containing the target genes and their flanking regions (including its native promoter and start codon) was amplified using the primers listed in Appendix A. The relaxed plasmid pSCT32-*gfp* (containing reporter gene *gfp* coding green fluorescent protein) was amplified with linearized primer pairs pSCT32-*gfp*-TL_F and pSCT32-*gfp*-TL_R (Appendix A), and the fragments were then inserted into plasmid pSCT32-*gfp* using a ClonExpress^®^ II One Step Cloning Kit (Vazyme Biotech Co., Ltd., Nanjing, China) to obtain a recombinant plasmid, which was transformed into GEB883-competent cells. The recombinant plasmids pSCT32-*gfp* with target regions were transferred into *V. alginolyticus* ZJ-T and ZJ-T-△*gcvB* by conjugation. The resulting strains were confirmed by PCR analysis and sequencing.

After fusions have been engineered on plasmids, Western-blotting was used for the quantification of these target protein fusions. For Western blotting, samples were harvested at OD600 of 0.3~0.4, 1.0~1.5, and 3.0~4.0 from LBS plus 5 μg/mL Cm (Chloramphenicol). Cells from three biological replicates were mixed together, then were centrifuged and resuspended in 100μL/OD_600_ of 2 × SDS loading buffer (Sangon Biotech, Shanghai, China), followed by incubation at 100 °C for 10 min. The proteins were separated by SDS-PAGE, and transferred to 0.2 μm polyvinylidene difluoride (PVDF) membranes (Millipore, MA, USA). The fused protein was detected using monoclonal anti-GFP (Sangon Biotech, Shanghai, China) with Dnak detected by polyclonal anti-Dnak (Abcam, Cambridge, UK) as loading control. SuperSignal™ West Pico PLUS (Thermo Fisher scientific, MA, USA) was used for visualization. The data presented are the most representative results of the three technical repetitions.

### 4.11. Hfq Recombinant Protein Construction and Purification

The full-length encoding sequence of *hfq* gene were amplified by PCR with the primer sets *hfq*-ORF_F / *hfq*-ORF_R (Appendix A). The plasmid pET28b was amplified with linearized primer pairs pET28b_F/R, and the fragments were then inserted into plasmid pET28b with a ClonExpress^®^ II One Step Cloning Kit (Vazyme Biotech Co., Ltd., Nanjing, China) to obtain a recombinant plasmid, which was transformed into *E. coli* BL21 (DE3)-competent cells. Strain pET28b-Hfq/BL21 (DE3) was grown to an OD_600_ of approximately 0.6 at 37 °C with shaking. Hfq production induction and purification were carried out according to reference [65]. Purified Hfq was concentrated using an Amicon Ultra centrifuge tube (Millipore, MA, USA) and stored in PBS buffer.

### 4.12. RNA Electrophoretic Mobility Shift Assays (RNA-EMSA)

The RNA oligonucleotides for T7 in vitro transcription template, listed in Appendix A, were produced by Sangon Biotech (Shanghai, China). Ribo^TM^ RNAmax-T7 Biotinylated Transcription Kit (Guangzhou, China) was used for T7 in vitro transcription which was designed to have a single biotin molecule at the 5′ end. T7 high yield RNA transcription kit (Vazyme Biotech Co., Ltd., Nanjing, China) was used for T7 in vitro transcription which was designed to have a non-biotinylated molecule at the 5′ end. The Hfq-GcvB RNA EMSA samples were prepared using an RNA-EMSA kit (BersinBio, Guangzhou, China) according to the instructions of the manufacturer. Hfq-RNA complexes were resolved by electrophoresis through a 6.5% nondenaturing polyacrylamide gel, transferred to a positively charged nylon membrane (Beyotime, Shanghai, China), and subjected to UV cross-linking (150 mJ). The chemiluminescent RNA-EMSA kit was used to visualize the biotinylated RNA.

### 4.13. RNA Stability

RNA stability measurement of *gcvB* gene was performed in *V. alginolyticus* wild-type strain ZJ-T, *hfq* knockout strain ZJ-T-Δ*hfq*, as previously described [35]. Overnight cultures from a single colony were diluted 1:1000 into LB medium plus 2.5%NaCl (LBS). Cultures were grown to early log phase (OD600 = 0.5~0.6), and 200 μg/mL of rifampin was added to the culture to stop transcription. Cells were harvested immediately (t = 0) and at 4, 8, 16, and 64 min following the rifampin addition and RNA was then purified from the samples as described above and used to generate cDNA. The *gcvB* gene, along with control 16S rRNA, were detected by qRT-PCR. The percentage of each of the RNAs remaining at each time point was calculated relative to t = 0 (100%).

## Figures and Tables

**Figure 1 ijms-23-09399-f001:**
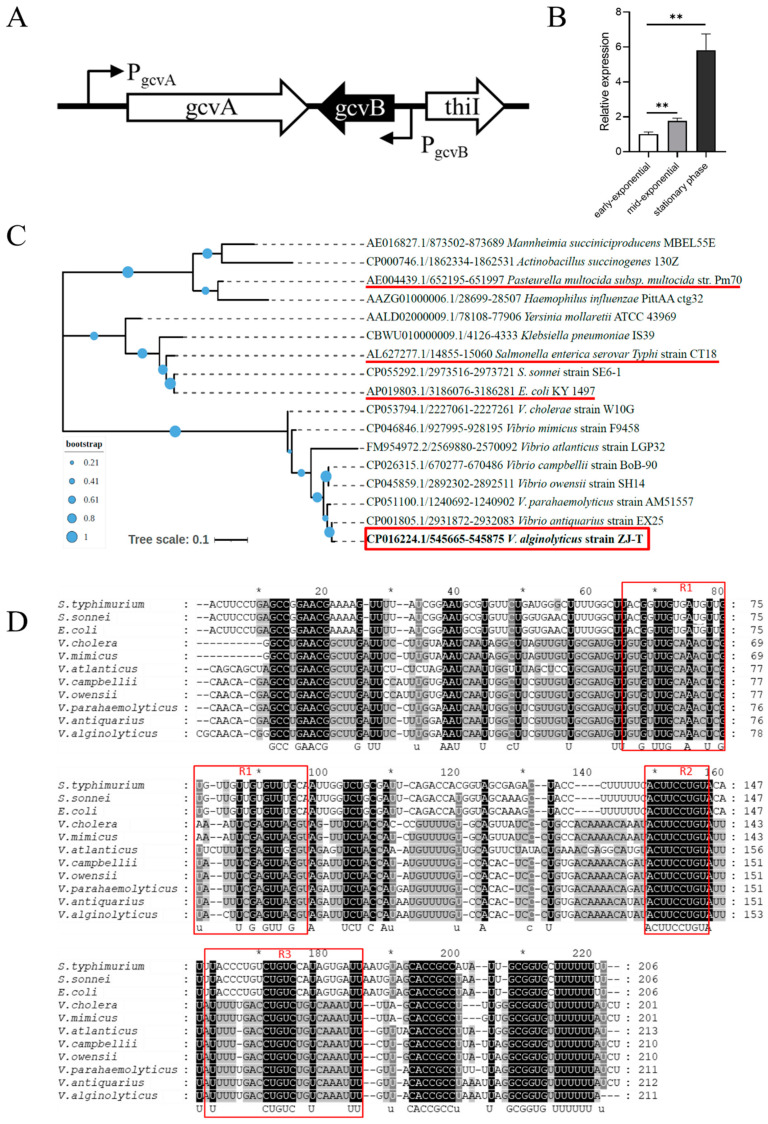
Bioinformatic analysis of *gcvB* sequence of *V. alginolyticus* and its expression in LBS. (**A**) Schematic diagram of small RNA GcvB encoding locations. P*_gcvA_* and P*_gcvB_* indicate gene *gcvA* and *gcvB* promoter, and arrows indicate transcription direction and coding direction; (**B**) phylogenetic analysis of *gcvB* among various species using the Maximum likelihood-method with a bootstrap value of 300; (**C**) conservation analysis of GcvB sequences among various species; (**D**) relative expression of GcvB in different phase (Student ’s *t*-test, *p* value: *, <0.05, **, <0.01). The levels of *gcvB* were normalized to the internal control 16S rRNA level. Error bars indicate standard deviations.

**Figure 2 ijms-23-09399-f002:**
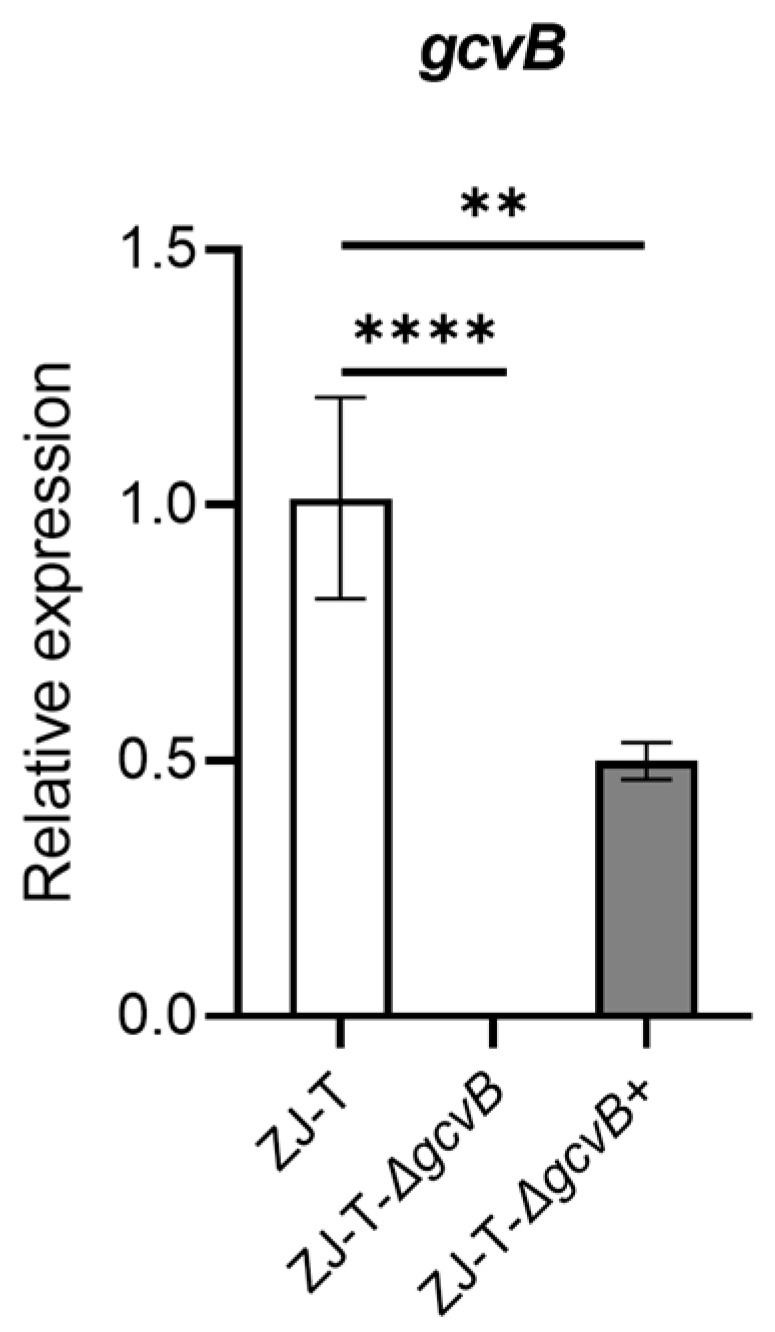
The relative expression of *gcvB* in wildtype, *gcvB* mutant and complementary strains (Student ’s *t*-test, *p* value: **, <0.01, ****, <0.0001). The levels of *gcvB* were normalized to the internal control 16S rRNA level. Error bars indicate standard deviations.

**Figure 3 ijms-23-09399-f003:**
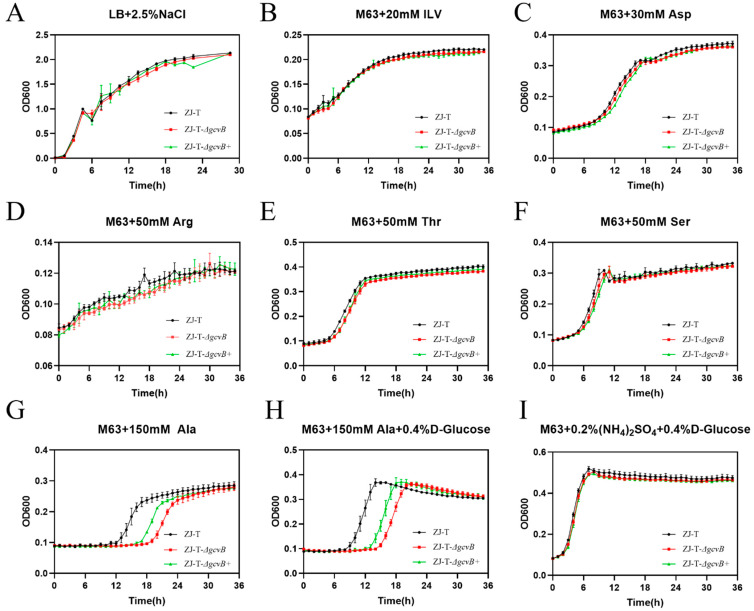
Growth curves of wildtype, *gcvB* knockout and complementary strains under different nutrient conditions. Growth curves of the wild type, *gcvB* knockout and complementary strains grown in LB + 2.5% NaCl rich medium(**A**), minimal medium M63 which D-glucose and (NH_4_)_2_SO_4_ were left out and replaced by the amino acids: branched chain amino acids (20 mM isoleucine, 20 mM leucine and 20 mM valine), (**B**) 30 mM L-Aspartic acid, (**C**) 50 mM L-Arginine, (**D**) 50 mM L-threonine, (**E**) 50 mM L-serine, (**F**) and 150 mM L-Alanine (**G**) as the sole carbon and nitrogen sources. Growth curves of M63 containing (NH_4_)_2_SO_4_ and 150 mM L-Alanine, (**H**) M63 plus (NH_4_)_2_SO_4_ plus D-glucose (**I**). For growth curves, three biological replicates are shown as points with their average values connected by lines. Error bars indicate the standard deviations (SD).

**Figure 4 ijms-23-09399-f004:**
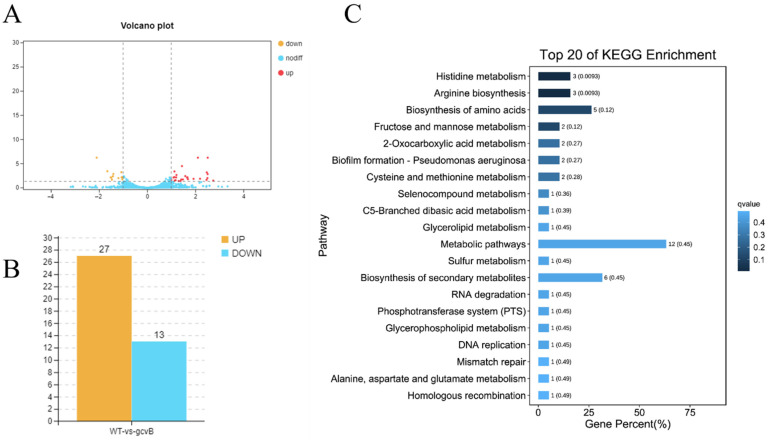
Overview of RNA transcriptomic profiles of wildtype and *gcvB* mutant strains. (**A**) Volcanic map of differential genes in transcriptome (FDR < 0.05, |log_2_(fold change)| ≥ 1). The red dot represents significantly up-regulated difference; the yellow dot represents significantly down-regulated difference; the blue dot represents no difference; (**B**) Statistical column chart of differential expressed genes. WT: wild-type strain *Vibrio alginolyticus* ZJ-T; *gcvB*: *gcvB* knockout strain ZJ-T-△*gcvB*; (**C**) histogram of top 20 of KEGG pathway enrichment in transcriptomics after inactivation of *gcvB*.

**Figure 5 ijms-23-09399-f005:**
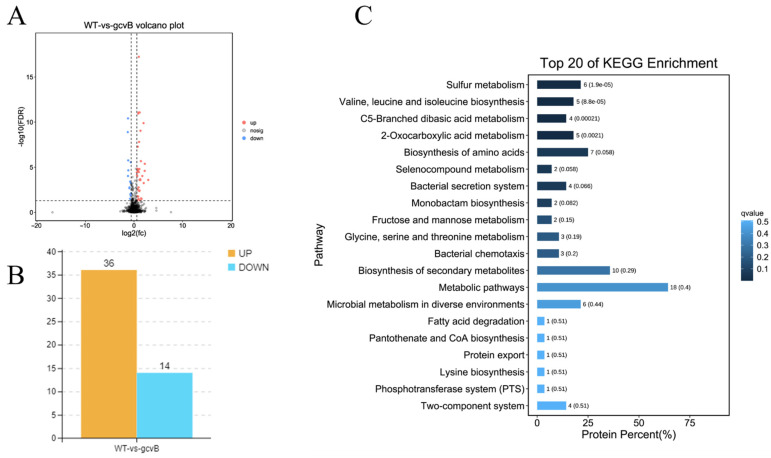
Overview of proteomic profiles of wildtype and *gcvB* mutant strains. (**A**) Volcanic map of differential genes in proteomics (FDR < 0.05, |fold change| ≥ 1.5). The red dot represents significantly up-regulated difference; the blue dot represents significantly down-regulated difference; the black dot represents no difference; (**B**) statistical column chart of differential expressed proteins. WT: wild-type strain *Vibrio alginolyticus* ZJ-T; *gcvB*: *gcvB* knockout strain ZJ-T-△*gcvB*; (**C**) histogram of top 20 of KEGG pathway enrichment in proteomics when *gcvB was knockout*.

**Figure 6 ijms-23-09399-f006:**
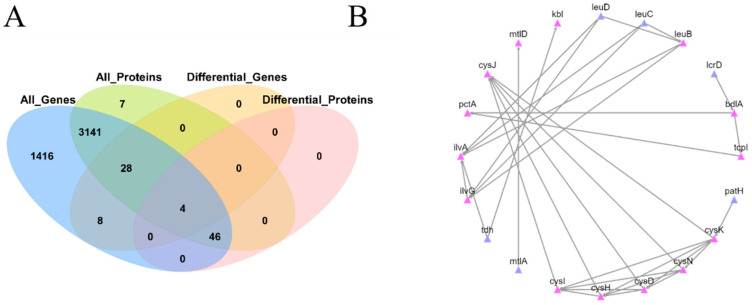
(**A**) Venn diagram of differentially expressed genes (DEGs) and differentially expressed proteins (DEPs) between wildtype and *gcvB* knockout strains; (**B**) PPI network of the identified DEPs (STRING score >  700). These genes showed consistent changes in gene expression with the shape of the lactation curve. The genes located at key nodes were also listed in Appendix A.

**Figure 7 ijms-23-09399-f007:**
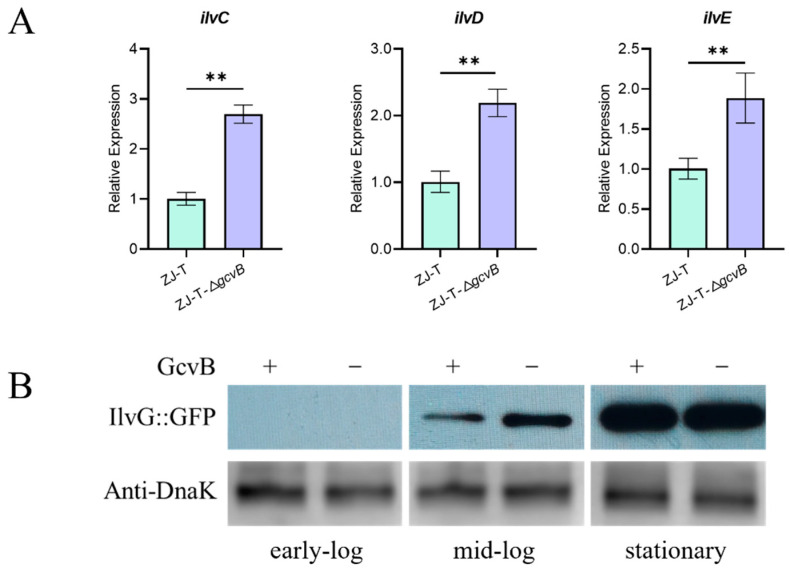
Effects of GcvB on valine/leucine/isoleucine biosynthetic pathway. (**A**) Relative expression of genes (*ilvC*, *ilvD*, *ilvE*) involved in the valine/leucine/isoleucine biosynthetic pathway by qRT-PCR (Student ’s *t*-test, *p* value: **, <0.01). The levels of *gcvB* were normalized to the internal control 16S rRNA level. Error bars indicate standard deviations; (**B**) Western blot detection of IlvG translational fusion. For Western blotting, samples were harvested at OD_600nm_ of 0.3~0.4 (early-log), 1.0~1.5 (mid-log), and 3.0~4.0 (stationary) from LBS plus 5 μg/mL Cm (chloramphenicol). Polyclonal anti-Dnak was used as loading control.

**Figure 8 ijms-23-09399-f008:**
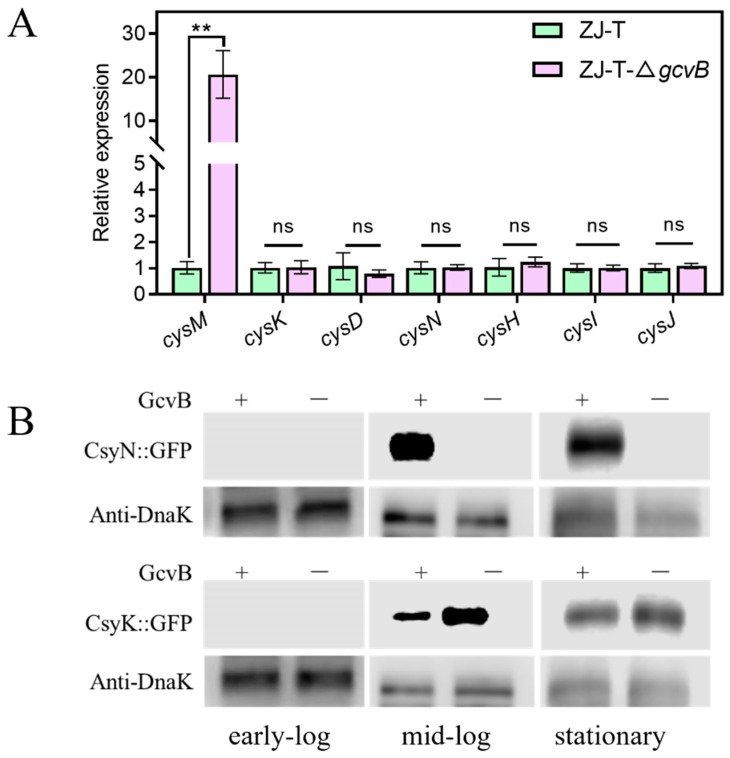
Effects of GcvB on sulfur and cysteine biosynthesis metabolism pathway. (**A**) Relative expression of genes involved in the sulfur and cysteine biosynthesis metabolism pathway by qRT-PCR (Student ’s *t*-test, *p* value: ns, >0.05, **, <0.01). The levels of *gcvB* were normalized to the internal control 16S rRNA level. Error bars indicate standard deviations; (**B**) Western blot detection of CysN and CysK translational fusion. For Western blotting, samples were harvested at OD600 of 0.3~0.4 (early-log), 1.0~1.5 (mid-log) and 3.0~4.0 (stationary) from LBS plus 5 μg/mL Cm (chloramphenicol). Polyclonal anti-Dnak was used as loading control.

**Figure 9 ijms-23-09399-f009:**
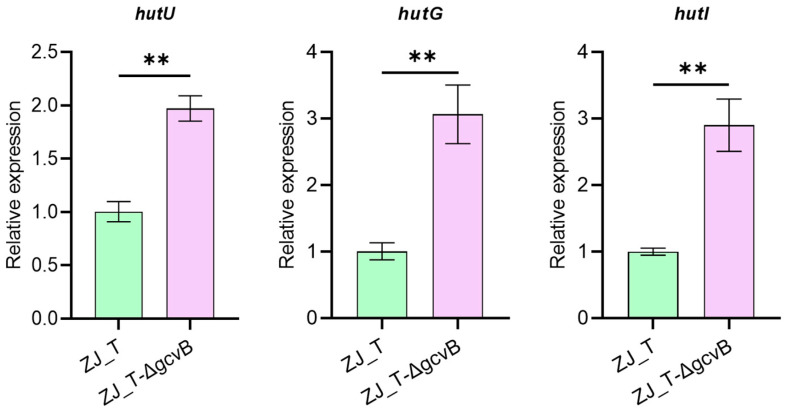
Relative expression of genes involved in the histidine metabolism pathway by qRT-PCR (Student ’s *t*-test, *p* value: **, <0.01). The levels of *gcvB* were normalized to the internal control 16S rRNA level. Error bars indicate standard deviations.

**Figure 10 ijms-23-09399-f010:**
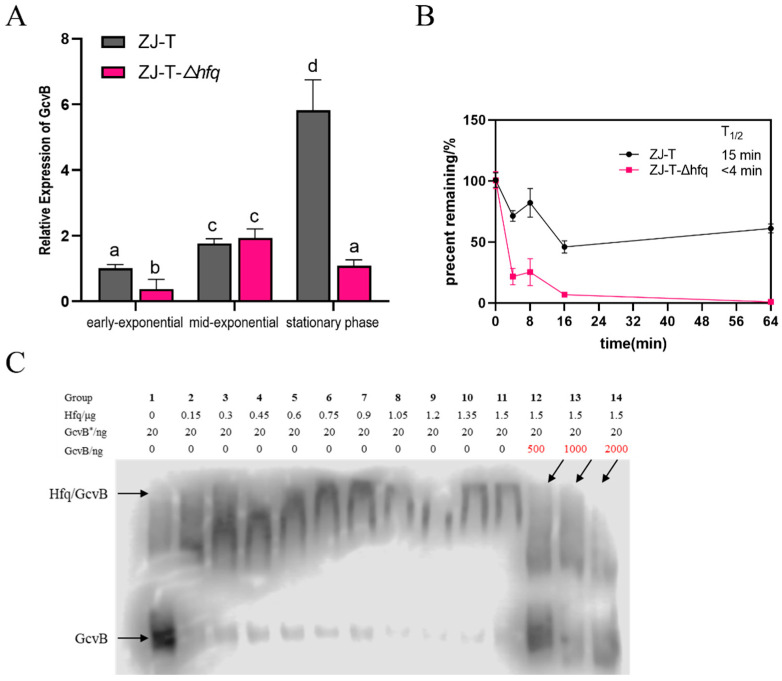
Effects of Hfq on GcvB. (**A**) Relative expression of GcvB at different periods of wildtype and *hfq* deletion strains. (**B**) Stability measurements of GcvB in hfq mutant strains. The half-life of GcvB is about 15 min in wild strain ZJ-T, but it is shortened to no more than 4 min in *hfq* deletion strain. (**C**) The Hfq protein binds specifically to GcvB. The picture shows the binding of GcvB to protein Hfq detected by RNA-EMSA. The Hfq protein was continuously added from 0 μg (group 1) to 1.5 μg (group 11). When the concentration of Hfq protein reached 1.05 μg (group 8), the binding of GcvB had reached saturation. After the addition of the non-biotinylated GcvB probe (groups 12, 13, 14), the biotinylated GcvB* was competitively eluted.

**Figure 11 ijms-23-09399-f011:**
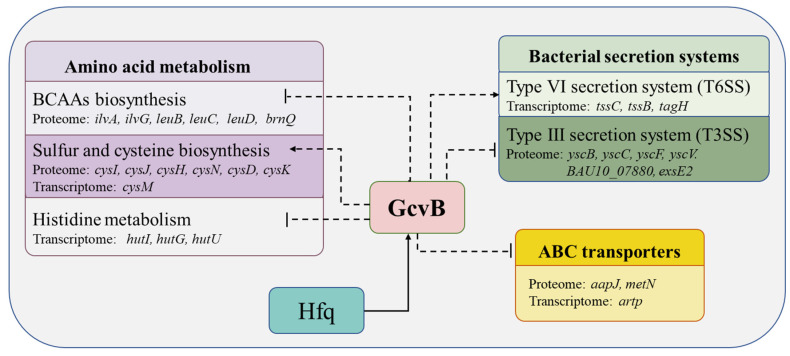
*V. alginolyticus* GcvB regulon. Boxes indicate major pathways regulated by GcvB. Solid arrows and bars indicate those that have been confirmed by EMSA. Dotted lines and bars show processes that have been confirmed by genetic or phenotypic analysis.

**Table 1 ijms-23-09399-t001:** Strains and plasmids used in this study.

Strains or Plasmids	Relevant Characteristics	Source
*Vibrio alginolyticus*		
ZJ-T	Ap^r^, translucent/smooth variant of wild strain ZJ51; isolated from diseased *Epinephelus coioides* off the Southern China coast	[55]
ZJ-T-Δ*gcvB*	Ap^r^; ZJ-T carrying a deletion of *gcvB*	This study
ZJ-T-△*gcvB^+^*	Cm^r^; ZJ-T carrying a GcvB complementation plasmid pMMB207-*gcvB*	This study
ZJ-T-Δ*hfq*	Ap^r^; ZJ-T carrying a deletion of *hfq*	[34]
ZJ-T/pSCT32-gfp-*ilvG*-TL	Cm^r^; ZJ-T carrying a *cysK* translational fusion plasmid pSCT32-gfp-*ilvG*-TL	This study
ZJ-T-△*gcvB*/pSCT32-gfp-*ilvG*-TL	Cm^r^; ZJ-T-△*gcvB* carrying a *cysK* translational fusion plasmid pSCT32-gfp-*ilvG*-TL	This study
ZJ-T/pSCT32-gfp-*cysK*-TL	Cm^r^; ZJ-T carrying a *cysK* translational fusion plasmid pSCT32-gfp-*cysK*-TL	This study
ZJ-T-△*gcvB*/pSCT32-gfp-*cysK*-TL	Cm^r^; ZJ-T-△*gcvB* carrying a *cysK* translational fusion plasmid pSCT32-gfp-*cysK*-TL	This study
ZJ-T/pSCT32-gfp-*cysN* -TL	Cm^r^; ZJ-T carrying a *cysN* translational fusion plasmid pSCT32-gfp-*cysD*-TL	This study
ZJ-T-△*gcvB*/SCT32-gfp-*cysN*-TL	Cm^r^; ZJ-T-△*gcvB* carrying a *cysN* translational fusion plasmid pSCT32-gfp-*cysD*-TL	This study
*E. coli*		
GEB883	WT; *E. coli* K12 *ΔdapA::ermpir* RP4-2 Δ*recA gyrA462*, *zei*298::Tn10; donor strain for conjugation	[56]
pET28b-Hfq/BL21(DE3)	Kan^r^; *E. coli* BL21(DE3) carrying the fusion expression plasmid pET28b-Hfq::His tag	This study
Plasmids		
pSW7848	Cm^r^; suicide vector with an R6K origin, requiring the Pir protein for its replication, and the *ccdB* toxin gene	[57]
pSW7848-Δ*gcvB*	Cm^r^; pSW7848 containing the mutant allele of Δ*gcvB*	This study
pMMB207	Cm^r^; RSF1010 derivative, *IncQ lacI*^q^ P*tac oriT*	[58]
pMMB207-*gcvB*	Cm^r^; pMMB207 containing the wild-type allele of *gcvB*	This study
pSCT32	Cm^r^; expression plasmid with a pBR322 and a f1 origin at the same time and a tac promoter	[59]
pSCT32-*gfp*	Cm^r^; pSCT32 containing reporter gene *gfp* coding green fluorescent protein	This study
pSCT32-gfp-*ilvG*-TL	Cm^r^; *ilvG* sequences (including its promotor and start codon) are translationally fused to pSCT32-gfp	This study
pSCT32-gfp-*cysK*-TL	Cm^r^; *cysK* sequences (including its promotor and start codon) are translationally fused to pSCT32-gfp	This study
pSCT32-gfp-*cysN*-TL	Cm^r^; *cysN* sequences (including its promotor and start codon) are translationally fused to pSCT32-gfp	This study
pET28b	Kan^r^; expression plasmid with a pBR322 origin, T7 promoter and 6×histag.	Xiaoxue Wang

Note: Cm^r^ and Ap^r^ indicate chloramphenicol and ampicillin resistance, respectively.

## Data Availability

All RNA sequencing data are deposited in the GenBank (wildtype biosample: SAMN29675353, SRA: SRR20124473-SRR20124475; *gcvB* mutant strain biosample: SAMN29675354, SRA: SRR20124470-SRR20124472) (accessed on 14 July 2022). All proteome data are deposited in iProX (Integrated Proteome Resources) database (https://www.iprox.cn/page/home.html, accessed on 19 July 2022), the accession ID is IPX0004724001. The other data presented in this study are available in the article.

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
