# Peer review of "GcvB Regulon Revealed by Transcriptomic and Proteomic Analysis in Vibrio alginolyticus"

_ijms, 2022, doi:10.3390/ijms23169399_

Round 1

Reviewer 1 Report

In the article entitled “GcvB regulon revealed by transcriptomic and proteomic analysis in Vibrio alginolyticus”, Liu et.al, investigated the role of GcvB, a conserved small RNA in V. alginolyticus using proteome and transcriptome approaches. Comparison of transcriptome and proteome data between wild-type ZJ-T and mutant strain suggests that GcvB affects the expression of <1% transcriptome and 1.52% proteome in V. alginolyticus. Characterization of GcvB in V. alginolyticus ZJ-T revealed 40 differentially expressed genes, while 50 genes displayed differences in protein products between wild-type ZJ-T and mutant strain. Most of the differentially expressed genes were involved in amino acid biosynthesis-transport and secretion systems. Current research will definitely enhance our understanding of the role of GcvB among the members of the family Vibrionacea.

The experiment is designed properly and the article is well written- easy to read and interpret. I do not have any major comments.

Author Response

Dear reviewers:

Re: Manuscript ID: ijms-1845653 and Title: GcvB regulon revealed by transcriptomic and proteomic analysis in Vibrio alginolyticus.

Thank you for your letter and the reviewers’ comments concerning our manuscript entitled “GcvB regulon revealed by transcriptomic and proteomic analysis in Vibrio alginolyticus” (ijms-1845653). Those comments are valuable and very helpful. We have read through comments carefully and have made corrections. Based on the instructions provided in your letter, we uploaded the file of the revised manuscript. Revisions in the text are shown using Yellow highlight. The responses to the reviewer's comments are marked in red and presented following.

We would love to thank you for allowing us to resubmit a revised copy of the manuscript and we highly appreciate your time and consideration. Should you have any questions, please contact us without hesitation.

Sincerely.

Bing LIU

Reviewer #1:

In the article entitled “GcvB regulon revealed by transcriptomic and proteomic analysis in Vibrio alginolyticus”, Liu et.al, investigated the role of GcvB, a conserved small RNA in V. alginolyticus using proteome and transcriptome approaches. Comparison of transcriptome and proteome data between wild-type ZJ-T and mutant strain suggests that GcvB affects the expression of <1% transcriptome and 1.52% proteome in V. alginolyticus. Characterization of GcvB in V. alginolyticus ZJ-T revealed 40 differentially expressed genes, while 50 genes displayed differences in protein products between wild-type ZJ-T and mutant strain. Most of the differentially expressed genes were involved in amino acid biosynthesis-transport and secretion systems. Current research will definitely enhance our understanding of the role of GcvB among the members of the family Vibrionacea.

The experiment is designed properly and the article is well written- easy to read and interpret. I do not have any major comments.

Response 1: Thank you for your comments concerning our manuscript.

Reviewer 2 Report

The manuscript (MS) deals with the understanding of the physiological role of GcvB and its mRNA targets in the Vibrionacea family using transcriptomic and proteomic approach. Although the proposed subject is interesting, the presented data have some limitations that should be taken into account prior publication.

Introduction     
-
The Introduction section should be improved. There is a lack of information about the novelity of the presented data. Which hypotheses/hypothesis was tested in the experiments?
-
Line 67 – the sentence “However, except Silveira et al reported that GcvB homologs were……” is not fully understood. Please improve it as it sounds to be very important. As I suppose, it indicates that some authors had received quite different results than others? Please describe these results in last paragraph of the introduction section. Furthermore, there is a lack of a number from the reference section (Silveira et al ……..) 

Results
- Line 82 – the name of the bacterial family should be written in italics - please revise the whole text.
-
Line 136 – delete capital letter for “Transcriptome”
-
Line 142 -  “RNAseq was done……” – this information should not be repeated, it can be found in the Materials and Methods section.
- Line 143 – it is not clear what “each group” means

Materials and Methods
- Line 381 - should be “Lysogeny Broth”

- The authors mentioned that “…….3 replicates in each case and three repetitions of the experiment were done….”. However it is not clear how many replicates the authors used to conduct other analyses such transcriptomic, proteomic, Western Blot

- The integrative analysis of transcriptomic and proteomic data should be conducted using appropriate bioinformatics tools such as Cytoscape or others.

In the introduction the authors said “…..we here report the characterization of gcvB and identification of its regulon by integrating the high-resolution RNA-seq and DIA assays…..”, however some tools that help to integrate these data were not used. In the results section the authors did not even visualized complex networks using the obtained valuable data. Therefore, I strongly encourage the authors to conduct the integrative analysis of the data.

Author Response

Dear reviewers:

Re: Manuscript ID: ijms-1845653 and Title: GcvB regulon revealed by transcriptomic and proteomic analysis in Vibrio alginolyticus.

Thank you for your letter and the reviewers’ comments concerning our manuscript entitled “GcvB regulon revealed by transcriptomic and proteomic analysis in Vibrio alginolyticus” (ijms-1845653). Those comments are valuable and very helpful. We have read through comments carefully and have made corrections. Based on the instructions provided in your letter, we uploaded the file of the revised manuscript. Revisions in the text are shown using Yellow highlight. The responses to the reviewer's comments are marked in red and presented following.

We would love to thank you for allowing us to resubmit a revised copy of the manuscript and we highly appreciate your time and consideration. Should you have any questions, please contact us without hesitation.

Sincerely.

Bing LIU

Reviewer #2:

Comments and Suggestions for Authors

The manuscript (MS) deals with the understanding of the physiological role of GcvB and its mRNA targets in the Vibrionacea family using transcriptomic and proteomic approach. Although the proposed subject is interesting, the presented data have some limitations that should be taken into account prior publication.

Introduction    

- The Introduction section should be improved. There is a lack of information about the novelity of the presented data. Which hypotheses/hypothesis was tested in the experiments?

Response 1: We are grateful for the suggestion. We have improved the Introduction section (from Line 68 to line 78)

- Line 67 – the sentence “However, except Silveira et al reported that GcvB homologs were……” is not fully understood. Please improve it as it sounds to be very important. As I suppose, it indicates that some authors had received quite different results than others? Please describe these results in last paragraph of the introduction section. Furthermore, there is a lack of a number from the reference section (Silveira et al ……..)

Response 2: We are grateful for the suggestion. .

Results

- Line 82 – the name of the bacterial family should be written in italics - please revise the whole text.

Response 3:We are grateful for the suggestion. The MS has been thoroughly read through and alld the names of the bacterial family have been written in italics .

- Line 136 – delete capital letter for “Transcriptome”

Response 4: We have revised it.

- Line 142 -  “RNAseq was done……” – this information should not be repeated, it can be found in the Materials and Methods section.

Response 5: We are grateful for the suggestion. We have changed it to “According to the data of transcriptome,” in line 148.

- Line 143 – it is not clear what “each group” means

Response 6: We are grateful for the suggestion. “Each group” means the wild type strain ZJ-T and gcvB knockout strain ZJ-T-ΔgcvB, which is shown now in line 149.

Materials and Methods

- Line 381 - should be “Lysogeny Broth”

Response 7: We are grateful for the suggestion. LB (Lysogeny broth, also known as Luria-Bertani) is the most commonly used growth medium to culture members of the Enterobacteriaceae. Therefore, we think two full names of LB to be OK. We have revised it to “Lysogeny Broth (LB)” in Line 401 according to your suggestion.

- The authors mentioned that “…….3 replicates in each case and three repetitions of the experiment were done….”. However it is not clear how many replicates the authors used to conduct other analyses such transcriptomic, proteomic, Western Blot

Response 8: We are grateful for the suggestion. Transcriptome and proteome are from the same sample; therefore, we have revised it as shown in the text from line 447 to line 450. For western blot,the details of the sample collection are shown in the text (line537, line 544).

- The integrative analysis of transcriptomic and proteomic data should be conducted using appropriate bioinformatics tools such as Cytoscape or others.

-  In the introduction the authors said “…..we here report the characterization of gcvB and identification of its regulon by integrating the high-resolution RNA-seq and DIA assays…..”, however some tools that help to integrate these data were not used. In the results section the authors did not even visualize complex networks using the obtained valuable data. Therefore, I strongly encourage the authors to conduct the integrative analysis of the data.

Response 9: Thank you for the suggestion. We conducted the interaction analysis and drew PPI network using Cytoscape (Figure 6). Besides, we summarized the regulatory role of GcvB with a diagram shown in Figure 11.

Round 2

Reviewer 2 Report

The manuscript has been imporved and could be accepted.

Author Response

Q: "The results showed that 40 genes are differentially expressed transcriptionally, 27 while 50 genes displayed differences in protein products between the wild type ZJ-T and gcvB mu-28 tant ZJ-T-ΔgcvB. Of which, only 4 genes showed difference both at the transcriptional level and protein level, indicating that most of the regulation occurred post-transcriptionally."
I do not understand the logic between the argumentation and the conclusion of the second sentence. It is more clearly explained in the result section lines 190-191:
"As many as 46 out of 50 DEPs (92%) showed no difference in transcription, indicating the majority of different expression of DEPs is the consequence of post-transcriptional 191 regulation by gcvB."
I find it necessary to clarify the sentence in the abstract.

 A: Thank you for your helpful suggestion. Indeed, this sentence has logical problems, we have modified it according to your suggestion, as shown below and also in lines 27-30 in the abstract.

“Transcriptome analysis revealed 40 genes differentially expressed (DEGs) between wild-type ZJ-T and gcvB mutant ZJ-T-ΔgcvB, while proteome analysis identified 50 differentially expressed proteins (DEPs) between them, but only 4 of them displayed transcriptional differences, indicating that most DEPs are the result of post-transcriptional regulation of gcvB.”